# Psychological Distress in Men during the COVID-19 Pandemic in Brazil: The Role of the Sociodemographic Variables, Uncertainty, and Social Support

**DOI:** 10.3390/ijerph19010350

**Published:** 2021-12-29

**Authors:** Anderson Reis de Sousa, Jules Ramon Brito Teixeira, Emanuel Missias Silva Palma, Wanderson Carneiro Moreira, Milena Bitencourt Santos, Herica Emilia Félix de Carvalho, Éric Santos Almeida, Raíssa Millena Silva Florencio, Aline Macêdo de Queiroz, Magno Conceição das Merces, Tilson Nunes Mota, Isabella Félix Meira Araújo, Josielson Costa da Silva, Sélton Diniz dos Santos, Emerson Lucas Silva Camargo, Luciano Garcia Lourenção, Richardson Augusto Rosendo da Silva, Evanilda Souza de Santana Carvalho, Iracema Lua, Sônia Barros, Tânia Maria de Araújo, Márcia Aparecida Ferreira de Oliveira, Álvaro Pereira, Wilson Abreu, Carlos Alberto da Cruz Sequeira

**Affiliations:** 1Escola de Enfermagem, Universidade Federal da Bahia, Salvador 40231-300, BA, Brazil; anderson.sousa@ufba.br (A.R.d.S.); milena_b.s@hotmail.com (M.B.S.); eriksdn@gmail.com (É.S.A.); isabellafelixmeira@hotmail.com (I.F.M.A.); josielson.silva@ufba.br (J.C.d.S.); alvaro_pereira_ba@yahoo.com.br (Á.P.); 2Departamento de Saúde, Universidade Estadual de Feira de Santana, Feira de Santana 44001-970, BA, Brazil; julesramon@gmail.com (J.R.B.T.); sdsantos@uefs.br (S.D.d.S.); evasscarvalho@uefs.br (E.S.d.S.C.); araujo.tania@uefs.br (T.M.d.A.); 3Escola Bahiana de Medicina e Saúde Pública, Salvador 40290-000, BA, Brazil; emanuelmssilva@gmail.com; 4Escola de Enfermagem, Universidade de São Paulo, Sao Paulo 05403-000, SP, Brazil; wanderson.moreira@usp.br (W.C.M.); sobarros@usp.br (S.B.); marciaap@usp.br (M.A.F.d.O.); 5Coordenação de Saúde do Adolescente, Jovem e Homem, Diretoria Técnica, Secretaria de Saúde, Prefeitura Municipal de Ananindeua, Ananindeua 67130-600, PA, Brazil; 6Escola de Enfermagem de Ribeirão Preto, Universidade de São Paulo, Ribeirao Preto 14040-902, SP, Brazil; hericacarvalho@usp.br; 7Faculdade Estácio de Alagoinhas, Alagoinhas 48010-970, BA, Brazil; raissaflorencio@yahoo.com.br; 8Faculdade de Enfermagem, Universidade Federal do Pará, Belem 66075-110, PA, Brazil; alinemacedo@ufpa.br; 9Departamento de Ciências da Vida, Universidade do Estado da Bahia, Salvador 1150-000, BA, Brazil; mmerces@uneb.br; 10Secretaria de Ciência, Tecnologia e Inovação, Salvador 41745-004, BA, Brazil; tilson.nunes.mota@gmail.com; 11Curso de Psicologia, Campus Ribeirão, Universidade de Ribeirão Preto, Ribeirao Preto 11440-003, SP, Brazil; lucmrg0@gmail.com; 12Escola de Enfermagem, Universidade Federal do Rio Grande, Rio Grande 96201-900, RS, Brazil; lucianolourencao.enf@gmail.com; 13Departamento de Enfermagem, Universidade Federal do Rio Grande do Norte, Natal 59012-300, RN, Brazil; rirosendo@hotmail.com; 14Instituto de Saúde Coletiva, Universidade Federal da Bahia, Salvador 40110-040, BA, Brazil; ira_lua@hotmail.com; 15Programa de Pós-Graduação em Enfermagem, Centro de Ciências da Saúde, Universidade Federal de Santa Maria, Santa Maria 97105-900, RS, Brazil; 16Centro de Investigação em Tecnologias e Serviços de Saúde, Escola Superior de Enfermagem do Porto, 4200-072 Porto, Portugal; wjabreu@esenf.pt

**Keywords:** men’s health, COVID-19, mental disorders, stress, psychological, social support

## Abstract

Objective: To analyze the relationships between sociodemographic variables, intolerance to uncertainty (INT), social support, and psychological distress (i.e., indicators of Common Mental Disorders (CMDs) and perceived stress (PS)) in Brazilian men during the COVID-19 pandemic. Methods: A cross-sectional study with national coverage, of the web survey type, and conducted with 1006 Brazilian men during the period of social circulation restriction imposed by the health authorities in Brazil for suppression of the coronavirus and control of the pandemic. Structural equation modeling analysis was performed. Results: Statistically significant direct effects of race/skin color (λ = 0.268; *p*-value < 0.001), socioeconomic status (SES) (λ = 0.306; *p*-value < 0.001), household composition (λ = 0.281; *p*-value < 0.001), PS (λ = 0.513; *p*-value < 0.001), and INT (λ = 0.421; *p*-value < 0.001) were evidenced in the occurrence of CMDs. Black-skinned men with higher SES, living alone, and with higher PS and INT levels presented higher prevalence values of CMDs. Conclusions: High levels of PS and INT were the factors that presented the strongest associations with the occurrence of CMDs among the men. It is necessary to implement actions to reduce the stress-generating sources as well as to promote an increase in resilience and the development of intrinsic reinforcements to deal with uncertain threats.

## 1. Introduction

The health crisis caused by COVID-19 exerted a negative impact on the mental health of the world’s population. The experience of unknown situations and the fear of contamination imposed by the burden of the disease and its repercussions produced new daily stressors and exacerbated previous ones already structurally present in societies [1,2,3,4]. Together, these factors generate continuous situations of suffering and contribute to the increase in psychological distress, as evidenced by indicators of Common Mental Disorders (CMDs) and perceived stress [5,6,7]. While the former refer to a set of symptoms, such as fatigue, insomnia, irritability, problems concentrating, and somatizations [5,6,7,8], the latter involves feelings about the uncontrollable and unpredictable nature of everyday events and the individual assessment of the ability to face such questions [4,5].

It is estimated that nearly one billion people are affected by some mental disorder due to the pandemic, with an emphasis on symptoms of anxiety, anguish and depression, and disorders resulting from alcohol use and substance abuse in addition to the increase in suicide rates [1,2,3,4,5,6]. The deleterious effects caused by the pandemic have been measured and show a reduction in life expectancy at birth, in the ability to work, and in absenteeism and presentism in addition to growth in unemployment and informality. Jointly, these situations install contexts that culminate in massive mental ailments in the male population [9,10,11,12,13].

The rapid spread of COVID-19, associated with the need for social distancing, forced quarantine, and blockades across the world, has placed imposed high stress levels on people [14,15], and sustaining these measures has increased the prevalence of CMDs in the populations [16,17], contributing to the increase in depressive and anxiety disorders, identified by the WHO as the most common psychiatric diseases in the world’s population [18,19]. Studies carried out prior to the COVID-19 pandemic identified high prevalences of CMD in men in Brazil, which ranged from 11.1% [20,21].

Consequently, the COVID-19 pandemic resulted in a psychologically chaotic and dismal setting. The radical changes in the individuals’ daily routines and the general context of fear and insecurity converge to unfamiliar situations and significant uncertainties that, due to their persistence over time, also provoke reactions of intolerance to this uncertainty [5,7]. Intolerance to uncertainty is described as the predisposition of an individual to consider the possibility of a negative event occurring regardless of the possibility of its actual occurrence as unacceptable [1,2].

Thus, intolerance to uncertainty is considered as one of the main underlying components of anxiety disorders [22], obsessive compulsive disorder (OCD) [23], post-traumatic stress disorder in depression [24], depression [25], and panic disorder [26]. Thus, inability to cope with uncertainty can be a negative predictor for well-being [27].

Social support is a basic human need and an important moderator of the impacts of the high psychological demands related to the COVID-19 pandemic. High social support has a protective effect and mitigates the negative impacts of stress on physical and psychological well-being [28]. Thus, the social support received during the pandemic is able to provide positive reinforcement to deal with stressful situations and prevent the occurrence/worsening of CMD.

In addition, there is evidence that CMDs are associated with social inequalities, as individuals in unfavorable socioeconomic conditions (e.g., black individuals, low income, and low level of education) are in a situation of greater vulnerability due to the constant experience of the feeling of insecurity, hopelessness, and risk of violence [29,30,31]. These sensations were possibly exacerbated by the COVID-19 pandemic, which supports the need to verify this association in men in Brazil, a country known to be marked by social inequalities and inequities.

Similarly, higher prevalence of CMD has been observed in people who live alone [32]. Overall, the loneliness of living alone largely explains the association with CMD [32]. Due to the need for social isolation to prevent and control COVID-19, people who live alone are even more susceptible to loneliness, which represents an increase in the potential risk for the development of CMD.

Epidemiological indicators, such as the number of new cases, the rates and length of hospital stay, the unfavorable outcomes of the disease, and the number of deaths have shown that the male population has been more impacted, evidencing that being a man is a risk factor for COVID-19 [33,34]. However, these results are restricted to the physical dimension of the disease and hardly advance the analysis of the mental health dimension [35,36].

In addition to that, population-based studies investigating the mental health situation in the context of the pandemic have included predominantly female samples [37,38,39], which limits a more comprehensive identification of the magnitude of the problem and, consequently, restricts the adoption of coping measures to a specific audience. In this context, this study contributes to overcoming this gap in scientific knowledge based on the research and appreciation of aspects related to the male population, which permeate the subjectivities and the social, symbolic, psychological, and psycho-emotional constructions of the population and their repercussions on health [40,41].

Given the above, it is considered crucial to give visibility to the mental health of male populations and to the aspects associated with the occurrence of mental disorders to strengthen the confrontation of the life dimensions affected by the pandemic in addition to contributing to overcoming this moment of profound health and social crisis generated by the dissemination of COVID-19.

Thus, this study aimed at analyzing the relationships between sociodemographic variables, intolerance to uncertainty (INT), social support, and psychological distress, that is, indicators of Common Mental Disorders (CMDs) and perceived stress (PS), in Brazilian men during the COVID-19 pandemic.

## 2. Methods

### 2.1. Type of Study

A cross-sectional study with national coverage, of the web survey type, and carried out during the period of social circulation restriction imposed by the health authorities in Brazil for suppression of the coronavirus and control of the pandemic.

### 2.2. Sample and Participants

The sample was estimated at 923 participants, considering the population of 64,520,660 Brazilian men with Internet access [42], 50% prevalence estimate, 95% confidence level, 5% precision, 80% power, effect of study design of two, and a 20% increase for losses.

The snowball technique [43] was used to recruit the participants through digital social networks (Facebook^®^, Instagram^®^, WhatsApp^®^, Grindr^®^). This is a non-probabilistic sampling technique performed by means of reference chains, where the first eligible and recruited participants invite new participants from their network of contacts (family, friends and acquaintances) who, in turn, indicate new participants and so on successively, until the estimated sample is minimally reached.

Initially, 25 participants were recruited, five from each of the Brazilian regions, who were called seeds. These participants were encouraged to send the research link to other men from their contact networks. At the end, 27 seeds were recorded, one from each of the Brazilian states.

The inclusion criteria adopted were as follows: being digitally literate to access the Internet and being at least 18 years old. The individuals excluded were those non-residents, in Brazil, or who were in the country at the time of data collection.

### 2.3. Procedures, Measurements, Variables and Outcome

For data collection, a questionnaire that was structured in blocks was used, including the following:

(a) Sociodemographic characteristics: sexual identity, age, schooling, self-reported race/skin color, house-sharing or not, work situation, and use of health plans;

(b) Common Mental Disorders (CMDs): the Self-Reporting Questionnaire (SRQ-20) was used to screen CMDs. SRQ-20 is validated for use in Brazil with satisfactory performance indicators [44,45,46]. It consists of 20 items with dichotomous answer categories (0—no; 1—yes). The cutoff point for men is at least five positive answers [47];

(c) Perceived Stress level: the ten-item version of the Perceived Stress Scale (PSS-10) [47], cross-culturally adapted [48] and validated for use in the Brazilian population [49], was used. The items have five-point Likert-type answer options (0 = never, 1 = almost never, 2 = sometimes, 3 = almost always, 4 = always) and are distinguished in questions with positive and negative connotations, thus inverting the score of the positive questions. The perceived stress score is calculated by the sum of the scores obtained in the ten items [47,48]. The levels were categorized as low (from zero to 13 points), moderate (from 14 to 26 points), and high (27 points or more) [50];

(d) Level of intolerance to uncertainty: the Intolerance of Uncertainty Scale (IUS) [51], 12-item version (IUS-12) [52], cross-culturally adapted [53] and validated for use in the Brazilian population [54], was applied. The items contain answer options arranged in a Likert scale, varying from one (not at all characteristics) to five (very characteristic) and assessing two dimensions: prospective IU (seven items) and inhibitory IU (five items). The validation study in Brazil evidenced a two-factor structure with high correlation between them (0.83) [55], which can indicate the existence of a higher-order factor (intolerance to uncertainty) [24,56]. The literature published to the present day indicates diverse evidence of the one-factor solution of intolerance to uncertainty as well as the use of its overall score [57,58]. The higher the score, the greater the level of intolerance to uncertainty. The IUS-12 latent construct showed adequate internal consistency (α = 0.89; ω = 0.89);

(e) Level of social support: the instrument used was the 2-Way Social Support Scale (2-WSSS) [58], 20-item version, transculturally adapted and validated in Brazil [59]. 2-WSSS assesses four social support dimensions: (1) receiving emotional support (seven items); (2) receiving instrumental support (four items); (3) offering emotional support (four items); and (4) offering instrumental support (five items). The items are answered in a Likert scale, varying from zero (it never applies) to five (it always applies). Higher scores indicate higher levels of social support. In this study, 2-WSSS presented satisfactory internal consistency (α = 0.85; ω = 0.96).

The self-reporting questionnaire was elaborated in a specific data-collection platform and made available through the Internet. From then onwards, it was widely publicized on Facebook^®^, Instagram^®^, WhatsApp^®^, and Grindr^®^ digital social networks by five trained researchers for autonomous and voluntary adherence as well as by making a direct invitation to the individuals who met the eligibility criteria. The data were collected between May and September 2020. The participants had access to the questionnaire after reading and signing the free and informed consent form and thus agreeing to participate in the study. Before each block of questions, the necessary information to answer them were made available as well as the option not to answer any question. The instructions about the recall period for each instrument were clarified.

### 2.4. Conceptual Framework and Study Hypotheses

Considering the evidence found in the literature, the understanding of human behaviors related to coping with the coronavirus outbreak and its impacts on people’s mental health, Directed Acyclic Graphs (DAGs) were constructed [60,61] to represent the conceptual structure of the common mental disorders in Brazilian men during the COVID-19 pandemic, with emphasis on the effect of race/skin color, house-sharing or not, social support, perceived stress, and intolerance to uncertainty. The directly observed variables are represented by rectangles and the latent variables by circles (Figure 1).

DAGs are diagrams that allow coding and explaining conceptual hypotheses [61], with growing recognition in the field of causal research in epidemiology [60,62,63]. In the DAGs, the relationships between events are represented by vertices connected by edges; the vertices represent the variables, and the edges show the possible ways or paths of relationships between variables, explaining causal links [61]. These causal paths can indicate direct causes if there is an arrow going from one variable to another or indirect causes if there is a sequence of arrows starting from one variable and reaching another, passing through one or more intermediate/mediating variables. Paths that do not follow the direction of the arrows linking exposure and outcome represent potential “confounding paths” [60].

### 2.5. Statistical Analysis

For the statistical analysis, the data were transferred from the Google Forms platform to a Microsoft Excel spreadsheet and were later organized in the SPSS software, version 24.0. Initially, a descriptive analysis of the variables of interest was performed for an estimation of frequencies (absolute and relative) and of prevalence values for CMDs. Pearson’s chi-square test was used, with a significance level of 5%, to verify the association between CMDs (outcome) and the independent variables. Subsequently, the database was exported to the Mplus software, version 8.0, for Structural Equation Modeling (SEM) analyses.

To perform the SEM, the latent constructs were measured (measuring models) [64]. To assess the factor structure of the observable items, Exploratory Factor Analysis (EFA) was performed, followed by Exploratory Structural Equation Models (ESEM) and Confirmatory Factor Analysis (CFA) to validate dimensionality of the construct elaborated [65], having as criteria a standardized factor load ≥ 0.3 and a residual variance ≤ 0.7 [66,67].

To evaluate the structural model constituted by the observed and latent variables, unadjusted and standardized regression coefficients were estimated, with 95% confidence intervals (95%CI) and *p*-value < 0.05. The size of the direct, indirect, and total effects were classified as follows: weak/small (around 0.10), moderate/medium (close to 0.30), and strong/big (>0.50) [68].

The Weighted Least-Squares Means and Variance Adjusted (WLSMV) estimator was used as a function of the modeling with categorical data. To re-specify the model, the Modification Indices (MI ≥ 10) and the Expected Parameter Changes (EPC ≥ 0.25) were assessed [69]. To assess fit of the models, the Root Mean-Square Error of Approximation (RMSEA > 0.06—exceptionally < 0.08, with a 90% confidence interval below 0.08) [69], the Comparative Fit Index (CFI ≥ 0.95), and the Tucker–Lewis Index (TLI ≥ 0.95) were adopted [68].

### 2.6. Ethical Considerations

Ethical approval regarding this study was obtained from the institutional ethics committee (decision: 4,087,611; CAAE: 32889420.9.0000.5531). All the participants in this study were only included after informed consent had been obtained from them. All procedures performed in this study were compatible with the ethical standards of the institutional research committee and with those of the Declaration of Helsinki and its comparable ethical standards.

## 3. Results

The study participants were 1006 men. The predominant profile included non-heterosexuals (54.1%), aged from 29 to 39 years old (45.1%), higher education (73.8%), black-skinned individuals (59.2%), without a partner (67.3%), with monthly income up to two minimum wages (41.6%), living with family members/friends (76.6%), and workers (75.0%). A high proportion of people depending exclusively on SUS care and services was observed (41.9%) (Table 1).

The overall prevalence of CMDs was 54.3%. Higher prevalence of CMDs was observed among the youngest individuals (62.7%), those with lower schooling (55.3%), black-skinned (55.1%), without a partner (58.2%), and among those who lived alone (58.3%), did not work (59.4%), had high levels of stress (92.4%), intolerance to uncertainty (82.3%), and received low social support (67.2%). The moderate stress levels reached 60.9%, while 36.0% reported high levels of intolerance to uncertainty. Receiving high social support was mentioned by 36.9% and offering high social support by 39.5% (Table 1).

Socioeconomic status (SES), intolerance to uncertainty (INT), and social support (SS) were treated as latent constructs. In general, the factor loads of the measuring models were high and statistically significant. The factor loads of the SES latent construct were above 0.60. In this construct, the highest load was observed for the occupational situation (OCP) indicator (λ = 0.690), and the lowest was found in the use of health plan (HP) indicator (λ = 0.609) (Table 2).

As for the INT construct, the exploratory factor analysis revealed, based on the eigenvalues, a solution of one predominant factor (eigenvalues = 5.907) with a marked reduction for two factors (eigenvalue = 1283). In the ESEM, the correlation between prospective and inhibitory IU was 0.992, and only the one-factor solution presented satisfactory fit indices. For this one-factor model, the highest load was observed for the “uncertainty makes me vulnerable” item (I7) (λ = 0.754) and the lowest one for “it is necessary to think about the future to avoid surprises” (I4) (λ = 0.393) (Table 2).

The SS measuring model was initially evaluated by the indicators of the first-order factors: receiving emotional support (RES), receiving instrumental support (RIS), offering emotional support (OES), and offering instrumental support (OIS), all with high factor loads and statistically significant. For RES and RIS, the highest loads were observed for the items “when I’m feeling down, there’s someone I can count on” (S3) (λ = 0.907) and “if I’m in trouble someone will help me” (S13) (λ = 0.845) and the lowest ones for “I feel I have a network of people who value me” (S7) (λ = 0.609) and “there’s someone who can help me fulfill my responsibilities” (S16) (λ = 0.594), respectively. For GES and GIS, greater burdens were verified for “I helped someone with their responsibilities when they were not able to fulfill them” (S18) (λ = 0.700) and “I comfort other people in difficult times” (S11) (λ = 0.867) and lower ones for “I’m a person whom others ask for help with tasks” (S20) (λ = 0.401) and “I’m a person available to listen to others’ problems” (S8) (λ = 0.697), respectively. The high correlation between the constructs of receiving (r = 0.832) and offering (r = 0.864) support endorsed the existence of the respective second-order factors called receiving support (RS) and offering support (OS), without a significant residual correlation (r = 0.430) (Table 2).

The SES, INT, and SS measuring models obtained satisfactory fit indices. The evaluation of the correlations between these latent constructs, both between the second-order factors of the SS model and in the model considering the correlations between all the latent variables, evidenced adequate discriminant validity (r < 0.90) (Table 3).

In the structural model, CMDs was considered as response variable. The SES, RS, OS, and INT latent variables and the self-reported race/color (COL), house-sharing (HS), and perceived stress (PS) observed variables were used as explanatory variables. The direct effects for CMDs were assessed for all the variables of the model, with the exception of OS, and their structural inter-relationships were considered in the paths of specific indirect effects. The estimated structural equation model presented adequate fit indices (Figure 2).

Statistically significant direct effects were evidenced with COL (λ = −0.268; *p*-value < 0.001), SES (λ = 0.306; *p*-value < 0.001), HS (λ = 0.281; *p*-value < 0.001), PS (λ = 0.513; *p*-value < 0.001), and INT (λ = 0.421; *p*-value < 0.001) for the CMDs. Thus, black-skinned men, those with higher socioeconomic status, living alone, and with higher levels of stress and intolerance to uncertainty presented higher prevalence values of CMDs. There were strong effects of PS and INT and medium effects of the other variables, showing that high levels of stress and intolerance to uncertainty were the factors with the strongest direct associations with the prevalence of CMDs (Figure 2).

The analysis of the specific indirect paths enabled the identification of important mediators of the effect of the explanatory variables on CMDs. Socioeconomic status was an important mediator of the relationship between race/skin color and CMDs, pointing out that non-black-skinned men with higher socioeconomic status presented a higher prevalence of CMDs. The effect of race/skin color and socioeconomic status was strong and significant (λ = 0.441; *p*-value < 0.001) (Figure 2).

The level of perceived stress was also a mediator in the association chain between SES and social support with CMDs. It was evidenced that men with lower socioeconomic status (λ = 0.349; *p*-value < 0.001), non-black-skinned (λ = 0.263; *p*-value < 0.001), who lived alone (λ = 0.547; *p*-value < 0.001), received low social support (λ = 0.860; *p*-value < 0.001), and offered high social support (λ = 0.536; *p*-value < 0.001) had higher stress levels; and higher stress level was associated with a higher prevalence of CMDs. The factors that most contributed (strong effects) to the increase in the level of perceived stress were the following: receiving low social support, living alone, and offering high social support (Figure 2).

Living alone mediated the effect of SES on CMDs, evidencing that men with higher socioeconomic status had higher stress levels and more prevalence of CMDs. The level of intolerance to uncertainty was also an important effect mediator: men who lived with family members/friends (λ = 0.170; *p*-value = 0.021), notably non-black-skinned and those with higher socioeconomic status, had a higher level of intolerance to uncertainty and higher prevalence of CMDs. Intolerance to uncertainty also mediated the effect of perceived stress on CMDs, indicating that men with higher stress levels, especially those with lower socioeconomic status, non-black-skinned, who lived alone and with low levels of receiving social support, and high levels of offering social support, had higher levels of intolerance to uncertainty (λ = 0.592; *p*-value < 0.001), resulting in a higher prevalence of CMDs. The factor that most contributed to the increase in intolerance to uncertainty was the high level of perceived stress, with a strong and significant effect (Figure 2).

The greatest total effects in CMDs were observed for higher level of perceived stress (λ = 0.763; *p*-value < 0.001), receiving low social support (λ = 0.716; *p*-value < 0.001), living alone (λ = 0.587; *p*-value < 0.001), high intolerance to uncertainty (λ = 0.421; *p*-value < 0.001), and high social support (λ = 0.409; *p*-value < 0.001), all strong and significant. There was a small overall effect of socioeconomic status (λ = 0.183; *p*-value < 0.001), but the indirect effects, mediated by high levels of perceived stress and intolerance to uncertainty, were robust and significant (Table 4).

It is to be noted that there was no statistically significant direct effect for CMDs from receiving and offering social support. However, among the specific indirect paths, the greatest effects included receiving low social support (λ = 0.442; *p*-value < 0.001), with strong and significant magnitude, and offering high social support (λ = 0.275; *p*-value < 0.001), of medium size and significant, both mediated by high stress levels. The medium effects of living alone mediated by high stress level (λ = 0.281; *p*-value < 0.001), of high stress level mediated by high level of intolerance to uncertainty (λ = 0.249; *p*-value < 0.001), and of low social support mediated by high levels of stress and intolerance to uncertainty (λ = 0.214; *p*-value < 0.001) also stood out (Table 4).

## 4. Discussion

The study evidenced a high prevalence of CMDs in men, which corroborates other findings on the high rates of mental illness in the Brazilian population during the COVID-19 pandemic [70,71]. In the pandemic context, no studies analyzing CMDs were identified, hindering comparisons. However, high prevalence values of sadness/depression, anxiety/nervousness, worsening of sleep problems, and panic syndrome were observed in Brazil during the pandemic [70,71]. These feelings experienced by men can be evoked to understand the high prevalence of CMD among them in the pandemic context since the instrument used to assess these minor mental disorders allows the identification of symptoms of anxious and depressive behavior, decreased energy, somatic symptoms, and depressive mood [44,45,46].

In this study, the high prevalence of CMDs is more pronounced among black-skinned men, higher socioeconomic status, living alone, and higher levels of PS and INT, and these last two factors are the ones most strongly associated with CMDs. Although OS has not presented any direct effect, specific indirect effects, such as receiving low social support and offering high social support, both mediated by high stress levels, were associated with CMDs.

The socioeconomic differences seem to exert an effect on the development of CMDs although the underlying mechanisms of that association are not yet well understood. The effect of higher socioeconomic status on the higher occurrence of CMDs is apparently controversial to what is documented in the literature [72,73], but some aspects can be evoked to understand this finding in this context of the COVID-19 pandemic: (a) the study was carried out in a virtual environment, which contributed to the lower access of men with low socioeconomic status and possible underestimation of CMDs in this stratum; (b) men with higher socioeconomic status may have a better perception of their mental health status due to a better schooling level [72], reflecting higher CMDs rates in a self-reporting questionnaire; (c) due to the restrictions imposed by the COVID-19 pandemic, men with higher socioeconomic status may be experiencing, abruptly and suddenly, more financial deprivation related to business and work, in social relationships, and in their life dynamics, situations that are already part of the daily lives of those with lower socioeconomic status even before the pandemic, and due to the absence of intrinsic reinforcement, they end up in mental distress; and (d) greater fear of poverty is associated with higher stress and anxiety levels among people with higher socioeconomic status during the pandemic, and maintenance of these levels reflects in mental illness [74]. Thus, this finding needs to be researched longitudinally in order to explore the possibilities of the causal link.

The “living alone” factor mediated the effect of SES on the CMDs so that those with higher socioeconomic status presented higher stress levels and higher prevalence of CMDs. These two factors, mediated by the high PS and INT levels, presented considerable indirect effects for CMDs. Living with family members/friends was related to a higher level of intolerance to uncertainty and to more prevalence of CMDs. In this context, although measures of social distancing and quarantine of the population are important for reducing morbidity and mortality due to COVID-19, their effects on the health of the population are undeniable. Regarding mental health, feelings and emotions, such as boredom, loneliness, anger, and sadness, can emerge not only because of the precautionary and control measures imposed but also because of the perception of vulnerability to contagion and risk of illness, especially among those who live alone [75].

The PS and INT factors are very inter-related so that men with higher PS levels had higher INT levels, resulting in greater prevalence of CMDs. The PS level was influenced by several factors, such as lower socioeconomic status, non-black-skinned people, living alone, receiving low social support, and offering high social support. A study carried out with Brazilian parents also identified that PS was the variable most strongly related to the CMD symptoms, showing that the higher the level of perceived stress, the greater the suspicion of these disorders [76]. Several factors have been associated with the occurrence of stress during the pandemic, which include concern about physical health and precautionary and disease control measures [76], anger and confusion arising from quarantine and social distancing [75], income reduction [75,76,77], fear of infection [75], and being part of a risk group for COVID-19 [77].

Regarding the socioeconomic situation of the men, the excess of demands, the fear of income loss, and the holding of the capital power may have contributed to increased stress levels and, consequently, to a higher prevalence of CMDs. This association is an important warning sign to be included in the global public health agenda, in a commitment to promote actions to reduce stress, increase literacy in male mental health for the perception of stress, and self-management of mental health care [78].

INT is a vulnerability factor associated with the development and maintenance of mental disorders [79]. Uncertainty about the future is a potentially stressful condition [80], hence the strong association with the occurrence of CMDs. The study evidenced that INT is negatively associated with mental well-being, especially when mediated by loneliness and fear of COVID-19, which are feelings intensely experienced during the pandemic [81]. Thus, it is necessary to implement actions to promote increased resilience and development and/or strengthening of intrinsic reinforcements to deal with the impacts of regular exposure to uncertain threats.

It is also highlighted that, although there are groups at higher risk, COVID-19 ends up affecting different social segments, and men tend to adhere less to the care measures and to neglect their health [82]. The occurrence and extension of this scenario can be even more impactful when considering social support, whether from the perspective of receiving or offering emotional and instrumental support. Thus, it is indispensable to formulate and strengthen already existing actions to promote psychosocial and emotional support for men in an exercise of citizenship and health care promotion that strengthens socio-affective networks, enabling men to have someone to count on in critical moments like a pandemic.

In addition to the categories of psychological disorders such as stress, other structural categories need to be analyzed in the scenario of impacts on male mental health, such as race/skin color and territory (geographic location and area of residence), since our findings identified that black-skinned men, those with higher socioeconomic status, who lived alone, and with higher levels of stress and intolerance to uncertainty had a higher prevalence of CMDs. Thus, an intersectional perspective needs to be employed as a way of explaining the vulnerabilities, inequalities, inequities, and necropolitics that promote overlapping impacts on the health, quality of life, and well-being of men who belong to marginalized groups, an aspect confirmed in our study when we identified the disparity in economic status among non-black-skinned men.

Although knowledge about the disease is advanced, there are still many gaps. Commonly, guided behaviors are based on scientific evidence, protocols, and guidelines. However, the recommendations themselves present certain degree of uncertainty, which generates doubts and discomfort in the face of the unknown on the part of authorities, health professionals, and the population [83]. Given this scenario, it is necessary to accept the existence of these uncertainties, limitations, and incompleteness in current knowledge. Transparency regarding these issues is a strategy to deal with the unknown in a rational and ethical way. In addition, the relevance of providing information and guidelines as recommended by the health authorities is emphasized as a way to avoid excessive simplifications and the dissemination of easy answers if they do not exist. Thus, gaining trust becomes one of the greatest challenges and ends up fighting mistrust and fake news [84], which intensify the emergence of conspiracy theories, misinformation, and “infodemic” [85,86,87,88,89,90].

Our study adopted a cross-sectional design and a sampling technique with recruitment by non-probabilistic methods, which limit the causal inference and external validity of the results. In addition to that, participation of men with no access to the interview via the Internet was excluded, which may have over- or under-estimated the prevalence of CMDs in the strata evaluated.

## 5. Conclusions

The results of this study point out the high prevalence of CMDs in men in the pandemic context and reinforce the need to assess the factors that precede this disease and to replicate the study with women. In addition to that, they endorse the importance of constructing latent variables to measure subjective aspects since three important latent constructs were revealed to be better studied, namely socioeconomic status, intolerance to uncertainty, and social support.

In addition to that, the study allowed identifying a higher prevalence of CMDs among younger men, with lower schooling, black-skinned, without a partner, who lived alone, did not work, with high levels of stress and intolerance to uncertainty, and who received low social support. It was also evidenced that the high levels of stress and intolerance to uncertainty were the factors that presented the strongest direct associations with the occurrence of CMDs.

These findings indicate that stress and intolerance to uncertainty are important factors in understanding mental suffering among men. Dealing with unexpected situations, with the loss of close people and future life perspectives accentuates stress levels, with the consequence of the occurrence and/or worsening of CMD. This situation is even worse in the presence of receiving low social support from close people and/or the government, as this support has an important mitigating effect on mental suffering, being able to resolve the harmful impacts of high psychological demand and the low control over adverse situations.

Thus, the importance of developing mental health promotion policies and actions for male populations is highlighted, considering that intervening in the aspects associated with the occurrence of CMDs is crucial to facing the impacts of the COVID-19 pandemic in the different life dimensions.

## Figures and Tables

**Figure 1 ijerph-19-00350-f001:**
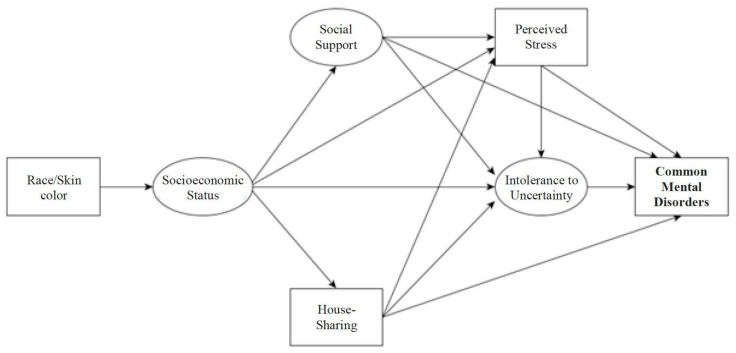
Conceptual structure of the determinants for the occurrence of CMDs in Brazilian men during the COVID-19 pandemic.

**Figure 2 ijerph-19-00350-f002:**
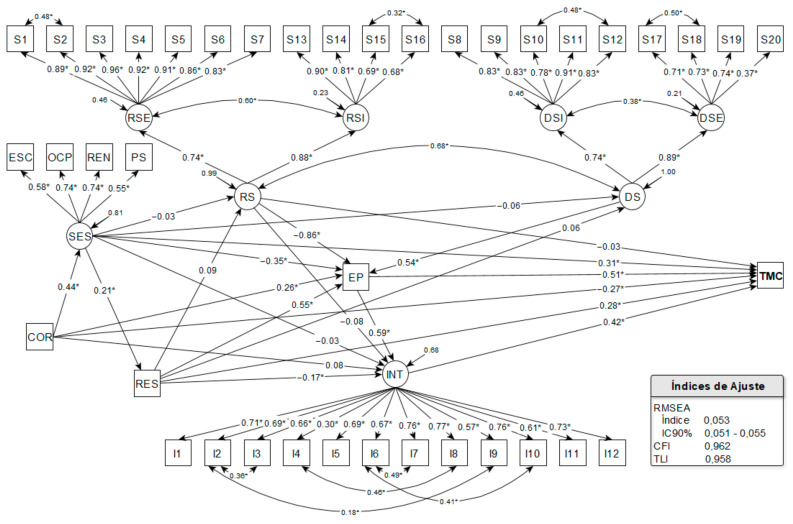
Structural equation model with direct and indirect specific effects for CMDs among Brazilian men in the COVID-19 pandemic context. Brazil, 2020. SES, socioeconomic status; INT, intolerance to uncertainty (I1–I12: indicating variables); RES, receiving emotional support (S1–S7: indicating variables); RIS, receiving instrumental support (S13–S16: indicating variables); OIS, offering instrumental support (S8–S12: indicating variables); OES, offering emotional support (S17–S20: indicating variables); RS, receiving support; OS, offering support; SCH, schooling; OCP, occupational status; INC, monthly income; HP, use of health plan; COL, race/skin color; HS, house-sharing; PS, perceived stress; I1–I12, items 1 to 12; S1–S20, items 1 to 20; RMSEA, Root Mean Square Error of Approximation; 90%CI, 90% confidence interval; CFI, Comparative Fit Index; TLI, Tucker–Lewis Index. * Statistically significant loads (*p*-value < 0.05).

**Table 1 ijerph-19-00350-t001:** Sociodemographic and occupational characteristics and prevalence of CMD in Brazilian men during the COVID-19 pandemic. Brazil, 2020. (N = 1.006).

Variables	*N*	%	N	*p*	*p*-Value
Sexual identity (N = 945)					
Heterosexual	434	45.9	228	52.5	0.785
Non-heterosexual	511	54.1	273	53.4	
Age group					
18 to 28 years old	314	31.2	197	62.7	<0.001
29 years old to 39 years old	454	45.1	257	56.6	
40 years or more	238	23.7	92	38.7	
Education					
Elementary/high school	264	26.2	146	55.3	0.696
University education	742	73.8	400	53.9	
Race/color (N = 1.001)					
White	376	37.4	197	52.4	0.247
Yellow	19	1.9	11	57.9	
Brown	397	39.5	211	53.1	
Black	196	19.5	116	59.2	
Indigenous	13	1.3	10	76.9	
Marital status					
With partner	329	32.7	152	46.2	<0.001
No partner	677	67.3	394	58.2	
Monthly income *					
Up to 2 salaries	418	41.6	203	56.4	0.083
3 to 4 salaries	228	22.7	109	47.8	
5 salaries or more	360	35.8	234	56.0	
Who they reside with (N = 1.005)					
Family/Friend(s)	770	76.6	409	53.1	0.163
Alone	235	23.4	137	59.3	
Work situation					
It works	755	75.0	397	52.6	0.062
Does not work	251	25.0	149	59.4	
Use of health plan					
Exclusively SUS	376	37.4	207	55.1	0.040
SUS and private plan	369	36.7	214	58.0	
Exclusively private plan	261	25.9	125	47.9	
Perceived stress					
Low	223	22.2	26	11.7	<0.001
Moderate	613	60.9	363	59.2	
High	170	16.9	157	92.4	
Uncertainty intolerance					
Low	327	32.5	79	24.2	<0.001
Moderate	317	31.5	169	53.3	
High	362	36.0	298	82.3	
Receive social support					
High	371	36.9	160	43.1	<0.001
Moderate	309	30.7	167	54.0	
Low	326	32.4	219	67.2	
Give social support					
Low	334	33.2	186	55.7	0.765
Moderate	275	27.3	145	52.7	
High	397	39.5	215	54.2	

*p*, prevalence; SUS, public health system. * Minimum wage in force in the period of data collection: R$ 1045,00.

**Table 2 ijerph-19-00350-t002:** Standardized factor loads of the measuring models of socioeconomic status, intolerance to uncertainty and social support among Brazilian men in the COVID-19 pandemic context. Brazil, 2020.

Latent Variables	Indicating Variables (Codes)	SFL ^a^	*p*-Value
SES			
Schooling (SCH)		0.631	<0.001
Occupational situation (OCP)		0.690	<0.001
Monthly income (INC)		0.664	<0.001
Use of health plan (HP)		0.609	<0.001
INT			
Uncertainty prevents me from living a full life (I1)		0.747	<0.001
I profoundly loathe unforeseen events (I2)		0.710	<0.001
I feel frustrated when I don’t have all the information I need (I3)		0.669	<0.001
It is necessary to think about the future to avoid surprises (I4)		0.393	<0.001
A small unforeseen event can ruin everything, even with the best planning (I5)		0.749	<0.001
When it’s time to act, uncertainty paralyzes me (I6)		0.620	<0.001
Uncertainty makes me vulnerable (I7)		0.754	<0.001
I always want to know what the future will bring me (I8)		0.611	<0.001
I hate to be taken by surprise (I9)		0.610	<0.001
The slightest sign of doubt dissuades me from acting (I10)		0.678	<0.001
I should be able to organize everything beforehand (I11)		0.634	<0.001
Uncertainty does not allow me to sleep well (I12)		0.694	<0.001
Receiving			
I have someone I can talk to about the pressures in my life (S1)		0.838	<0.001
There is at least one person with whom I can share most of the things (S2)		0.838	<0.001
When I’m feeling down, there’s someone I can count on (S3)		0.907	<0.001
I have someone in my life who offers me emotional support (S4)		0.899	<0.001
There is at least one person in whom I feel I can trust (S5)		0.820	<0.001
There is someone in my life who makes me feel that I’m worthy (S6)		0.689	<0.001
I feel that I have a network of people who value me (S7)		0.609	<0.001
RIS			
If I’m in trouble, someone will help me (S13)		0.845	<0.001
I have someone to help me when I’m ill (S14)		0.686	<0.001
If I need money, I know someone who can help me (S15)		0.649	<0.001
There is someone who can help me fulfill my responsibilities (S16)		0.594	<0.001
OIS			
I’m a person who is available to listen to others’ problems (S8)		0.697	<0.001
I seek to encourage people when they’re feeling down (S9)		0.723	<0.001
People close to me tell me their deepest concerns (S10)		0.710	<0.001
I comfort other people in difficult times (S11)		0.867	<0.001
People trust me when they have problems (S12)		0.737	<0.001
OES			
I help others when they are too busy (S17)		0.685	<0.001
I helped someone with their responsibilities when they were not able to fulfill them (S18)		0.700	<0.001
I provided help when someone who lived with me was ill (S19)		0.588	<0.001
I’m a person whom others ask for help with tasks (S20)		0.401	<0.001
RS ^a^			
RES ^b^		0.882	<0.001
RIS ^b^		0.949	<0.001
Offering			
OIS ^b^		0.857	<0.001
OES ^b^		0.740	<0.001

SFL, standardized factor loads; SES, socioeconomic status; INT, intolerance to uncertainty; RES, receiving emotional support; RIS, receiving instrumental support; OIS, offering instrumental support; OES, offering emotional support; RS, receiving support; OS, offering support. ^a^ 2nd-order factor; ^b^ 1st-order factor.

**Table 3 ijerph-19-00350-t003:** Fit indicators of the measuring models, using CMDs as response variable, Brazil, 2020.

Indices	SES	INT	SS (RS ▯ OS)	CMM
RMSEA				
Index	0.038	0.049	0.046	0.046
90% CI	0.000–0.078	0.040–0.059	0.041–0.050	0.043–0.048
*p*-Value	0.671	0.518	0.939	0.998
CFI	0.997	0.992	0.959	0.976
TLI	0.991	0.986	0.950	0.974
R ^a^				
RS ▯ OS	-	-	0.430	-
SES ▯ INT	-	-	-	−0.098
SES ▯ RS	-	-	-	−0.019
SES ▯ OS	-	-	-	−0.033
INT ▯ RS	-	-	-	−0.325
INT ▯ OS	-	-	-	−0.067

SES, socioeconomic status; INT, intolerance to uncertainty; SS, social support; RS, receiving social support; OS, offering social support; CMM, Correlated Measuring Models; RMSEA, Root Mean Square Error of Approximation; 90% CI, 90% confidence interval; TLI, Tucker–Lewis Index; CFI, Comparative Fit Index. ^a^ Residual correlations (1) between the latent variables.

**Table 4 ijerph-19-00350-t004:** Standardized total and indirect effects of the structural equation model, using CMDs as response variable among Brazilian men in the COVID-19 pandemic context. Brazil, 2020.

Paths	SFL	SE	Est/SE ^a^	*p*-Value ^b^
Total Effects				
INT → CMDs	0.421	0.047	8.87	<0.001
RS → CMDs	−0.716	0.051	−14.12	<0.001
OS → CMDs	0.409	0.021	19.11	<0.001
SES → CMDs	0.183	0.043	4.21	<0.001
PS → CMDs	0.763	0.039	19.46	<0.001
COL → CMDs	−0.021	0.040	−0.53	0.595
HS → CMDs	0.587	0.035	16.75	<0.001
Specific indirect effects				
RS				
RS → PS → CMDs	−0.442	0.039	−11.46	<0.001
RS → INT → CMDs	−0.032	0.025	−1.31	0.190
RS → PS → INT → CMDs	−0.214	0.037	−5.84	<0.001
OS				
OS → PS → CMDs	0.275	0.012	23.48	<0.001
OS → PS → INT → CMDs	0.134	0.022	5.97	<0.001
SES				
SES → HS → CMDs	0.060	0.014	4.22	<0.001
SES → PS → CMDs	−0.179	0.036	−4.96	<0.001
SES → INT → CMDs	0.021	0.027	0.79	0.429
SES → RS → CMDs	0.001	0.002	0.38	0.701
SES → HS → PS → CMDs	0.060	0.014	4.22	<0.001
SES → RS → PS → CMDs	0.012	0.025	0.48	0.629
SES → OS → PS → CMDs	−0.015	0.015	−1.00	0.318
SES → HS → INT → CMDs	−0.015	0.008	−1.88	0.060
SES → PS → INT → CMDs	−0.087	0.022	−3.96	<0.001
SES → RS → INT → CMDs	0.001	0.002	0.49	0.625
SES → HS → RS → CMDs	0.001	0.001	−0.42	0.673
SES → HS → RS → PS → CMDs	−0.008	0.006	−1.37	0.170
SES → HS → OS → PS → CMDs	0.003	0.003	0.93	0.351
SES → HS → PS → INT → CMDs	0.029	0.009	3.31	0.001
SES → RS → PS → INT → CMDs	0.006	0.012	0.48	0.629
SES → OS → PS → INT → CMDs	−0.007	0.008	−0.99	0.325
SES → HS → RS → INT → CMDs	−0.001	0.001	−1.06	0.289
SES → HS → RS → PS → INT → CMDs	−0.004	0.003	−1.37	0.171
SES → HS → OS → PS → INT → CMDs	0.002	0.002	0.92	0.356
PS				
PS → INT → CMDs	0.249	0.040	6.22	<0.001
COL				
COL → PS → CMDs	0.135	0.006	20.80	<0.001
COL → INT → CMDs	−0.034	0.019	−1.78	0.075
COL → SES → CMDs	0.135	0.006	20.80	<0.001
COL → SES → HS → CMDs	0.026	0.006	4.21	<0.001
COL → SES → PS → CMDs	−0.079	0.016	−4.93	<0.001
COL → PS → INT → CMDs	0.066	0.011	5.92	<0.001
COL → SES → INT → CMDs	0.009	0.012	0.79	0.429
COL → SES → RS → CMDs	0.000	0.001	0.38	0.702
COL → SES → HS → PS → CMDs	0.026	0.006	4.21	<0.001
COL → SES → RS → PS → CMDs	0.005	0.011	0.48	0.629
COL → SES → OS → PS → CMDs	−0.007	0.007	−1.00	0.319
COL → SES → HS → INT → CMDs	−0.007	0.004	−1.88	0.060
COL → SES → PS → INT → CMDs	−0.038	0.010	−3.95	<0.001
COL → SES → RS → INT → CMDs	0.000	0.001	0.49	0.625
COL → SES → HS → RS → CMDs	0.000	0.001	−0.42	0.673
COL → SES → HS → RS → PS → CMDs	−0.004	0.003	−1.37	0.170
COL → SES → HS → OS → PS → CMDs	0.001	0.002	0.93	0.351
COL → SES → HS → PS → INT → CMDs	0.013	0.004	3.30	0.001
COL → SES → RS → PS → INT → CMDs	0.003	0.005	0.48	0.629
COL → SES → OS → PS → INT → CMDs	−0.003	0.003	−0.99	0.325
COL → SES → HS → RS → INT → CMDs	0.000	0.000	−1.06	0.289
COL → SES → HS → RS → PS → INT → CMDs	−0.002	0.001	−1.37	0.171
COL → SES → HS → OS → EP → INT → CMDs	0.001	0.001	0.92	0.356
HS				
HS → PS → CMDs	0.281	0.012	23.79	<0.001
HS → INT → CMDs	−0.071	0.033	−2.17	0.030
HS → RS → CMDs	−0.002	0.006	−0.43	0.668
HS → RS → PS → CMDs	−0.038	0.026	−1.45	0.146
HS → OS → PS → CMDs	0.015	0.016	0.97	0.334
HS → PS → INT → CMDs	0.136	0.023	5.93	<0.001
HS → RS → INT → CMDs	−0.003	0.003	−1.10	0.273
HS → RS → PS → INT → CMDs	−0.019	0.013	−1.46	0.144
HS → OS → PS → INT → CMDs	0.007	0.008	0.96	0.338

SFL, standardized factor loads; SE, standard error; Est/SE, ratio between SFL estimate and SE; SES, socioeconomic status; INT, intolerance to uncertainty; RS, receiving support; OS, offering support; HS, house-sharing; COL, self-reported race/skin color; PS, perceived stress; CMDs, common mental disorders. ^a^ Statistically significant if −1.96 > Est/SE > 1.96. ^b^ Statistically significant if <0.05.

## Data Availability

The datasets generated during the current study are not publicly available but are available from the corresponding author on reasonable request.

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
