# Peer review of "Psychological Distress in Men during the COVID-19 Pandemic in Brazil: The Role of the Sociodemographic Variables, Uncertainty, and Social Support"

_ijerph, 2021, doi:10.3390/ijerph19010350_

Round 1
Reviewer 1 Report
Thank you for being able to review the manuscript,
The authors analyze the psychological discomfort in men during the COVID-19 pandemic in Brazil and the role of sociodemographic variables, uncertainty and social support the objective of the manuscript is to analyze the relationships between sociodemographic variables, intolerance
to uncertainty, social support and psychological anguish in Brazilian men during the COVID-19 pandemic, through a cross-sectional study with national coverage, web survey type, and carried out with 1,006 Brazilian men during the period of social circulation restriction imposed by health authorities in Brazil, for the suppression of the coronavirus and control of the pandemic. For this, a structural equation modeling analysis was carried out, with results of statistically significant direct effects of race / skin color, socioeconomic level, household composition, PS and INT. Black-skinned men, with higher SES, who lived alone and with higher levels of PS and INT presented higher prevalence values of CMD.
It's a good work, on a very interesting topic such as covid-19, but before being published it must incorporate the following suggestions, I ask the authors to follow them one by one,
Some important references are missing in the introduction, check it out,
The discussion section must be improved and more worked, much more worked, connecting the scientific literature with the introduction, that will give more power to the manuscript,
The same, in the conclusions section, it's very brief, there are hardly a few lines, which are insufficient for a work to be published in our magazine,
With these changes made, the article will improve and have the quality to be published in this prestigious journal,
kind regards
Author Response
-Some important references are missing in the introduction, check it out.
Authors' response: We appreciate this comment. In response we add new references.
-The discussion section must be improved and more worked, much more worked, connecting the scientific literature with the introduction, that will give more power to the manuscript.
Authors' response: We appreciate this comment. We believe that, with the suggestions indicated by all the reviewers about decreasing and adding to the discussion, this request was considered.
-The same, in the conclusions section, it's very brief, there are hardly a few lines, which are insufficient for a work to be published in our magazine,
Authors' response: We appreciate this comment. In response, we expand the conclusions section.
Reviewer 2 Report
From REVIEWER
First, let me say thank you for the opportunity to review this paper.
The article "Psychological distress in men during the COVID-19 pandemic in Brazil: The role of the sociodemographic variables, uncertainty and social support " points out the high prevalence of CMDs in men in the pandemic context and reinforce the need to assess the factors that precede this disease and to replicate the study with women. In addition to that, they endorse the importance of constructing latent variables to measure subjective aspects, since three important latent constructs were revealed to be better studied, namely: Socioeconomic Status, Intolerance to Uncertainty, and Social Support. The authors analyzed the relationships between sociodemographic variables, intolerance to uncertainty (INT), social support and psychological distress, that is, indicators of Common Mental Disorders (CMDs) and Perceived Stress (PS), in Brazilian men during the COVID-19 pandemic.
In my view, it is a strength of this study that they conducted a cross-sectional study with national coverage, of the web survey type and carried out during the period of social circulation restriction imposed by the health authorities in Brazil, for suppression of the coronavirus and control of the pandemic.
Overall, I have no qualms about the SEM methods presented. This manuscript is well-addressed and organized into the essential sections for publication. I would strongly recommend accepting this manuscript to our next journal issue.
Author Response
Thanks for the comment.
Reviewer 3 Report
Dear Authors,
This manuscript is well-structured, interesting, and employed sound methodology. Congratulations for your great work.
I have only several comments.
- The sentence in line 53-56 should be supported by any previous studies.
- Line 355 Authors mention that high prevalence of CMDs in this study sample. It is better to add some reference to compare the prevalence in this study.
- The sentence in line 419-420, said something obvious and some specific example is helpful for measure to maintain men's CMDs.
- Authors seemed to stress that intolerance to uncertainty is a vulnerability factor, but it was difficult to find any idea how to improve INT, which could help men maintain their mental health in post-COVID era. Authors also should clarify whether individuals' INT could be changed by any intervention, and add discussion regarding how it is applied following the pandemic.
Author Response
1. The sentence in line 53-56 should be supported by any previous studies.
Authors' response: Thanks for this comment, references have been added.
2. Line 355 Authors mention that high prevalence of CMDs in this study sample. It is better to add some reference to compare the prevalence in this study.
Authors' response: Thanks for this comment, references have been added.
3. The sentence in line 419-420, said something obvious and some specific example is helpful for measuring to maintain men's CMDs.
Authors' response: Thank you for this comment, we chose to delete the phrase.
4. Authors seemed to stress that intolerance to uncertainty is a vulnerability factor, but it was difficult to find any idea how to improve INT, which could help men maintain their mental health in post-COVID was. Authors also should clarify whether individuals' INT could be changed by any intervention, and add discussion regarding how it is applied following the pandemic.
Authors' response: We appreciate this comment, however, in the same paragraph where we highlight this vulnerability caused by intolerance to uncertainty, we also emphasize that it is necessary to promote increased resilience and the development and/or strengthening of intrinsic reinforcements to mitigate the impacts of exposure regulate uncertain threats, such as the COVID-19 pandemic.

Reviewer 4 Report
The topic of this paper is the psychological distress in men in Brazil during the Covid-19 pandemic. The paper objective is to analyze the relationships between sociodemographic variables, intolerance to uncertainty, social support and perceived stress and the scores at Common Mental Disorders questionnaire.
The background of the study needs to be developed, including also studies that use Common Mental Disorders questionnaire as screening instrument for mental disorder before the pandemic.
The results part will benefit if the authors add a table with the sociodemographic variable of the study sample.
Author Response
-The background of the study needs to be developed, including also studies that use the Common Mental Disorders Questionnaire as a screening instrument for mental disorder before the pandemic.
Authors' response: Thank you for this comment, we add results from studies evaluating CMD in Brazilian men prior to the pandemic.
-The results part will benefit if the authors add a table with the sociodemographic variable of the study sample.
Authors' response: Thank you for this comment, we have added table 1 with descriptive statistics of sociodemographic characteristics and levels of stress, intolerance to uncertainty, receiving and giving social support.

Reviewer 5 Report
The study covers important issues of the pathways of the pandemic impact on mental health.
However I recommend considering the following issues to enhance the presentation of the study background and results.
My specific comments are detailed below.
Introduction
The first 3 sentences need to be grounded in the literature, each one, since they bring different ideas.
Line 69: “The radical changes in the…” please support the statement with references.
Line 72: “Intolerance to uncertainty…” Please provide original references for this construct.
The introduction provides a good background for investigation of the construct of “social intolerance” in the context of common mental health issues during the pandemic. However, the literature on impact of additional factors, which are set to be the study variables, such as social support and specific socio-demographic data (race, house-sharing, etc) is missing. Please support with the previous research variables that were selected for the investigation.
Results
Please provide a table with descriptive statistics on demographic data and the study measurements. For example. It is not clear from your report who “youngest individuals” are or what “lower schooling” means.
Furthermore, the descriptive presentation of the data should be supported with statistical analysis. For example, χ2 may provide a good estimation whether the differences in the CMD prevalence between the young and older participants are not occasional.
Discussion
Line 355: Please provide a reference for the statement. What new information presents the following sentence starting with “high prevalence values of sadness/depression…” (Line 358)?
Lines 361-366: Please move this paragraph to the Introduction. Be oriented toward explanation of your findings.
In general, the discussion is too long with multiple repetitions on the main topics, such as vulnerability due to intolerance to uncertainty (lines 453-461) or cultural issues (Lines 472-477/ 373-379).
In addition, some pieces of information are not relevant for the study, even though they are generally interesting. For example, the statements on non-biological nature of mental health (“These results reinforce that mental health goes beyond the biological issues, …”) do not fit the study purposes and design since no biological measurements were done.
I recommend to revise and reorganize the discussion to bring the main information without duplications and with a focus on the research main concepts.
In general, the authors bring a lot of references, while part of information still remains ungrounded. Please revise the references and use the stronger ones.
Good luck!
Author Response
Introduction
-The first 3 sentences need to be grounded in the literature, each one, since they bring different ideas.
Authors' response: Thanks for this comment, references have been added.
-Line 69: “The radical changes in the…” please support the statement with references.
Authors' response: Thanks for this comment, references have been added.
-Line 72: “Intolerance to uncertainty…” Please provide original references for this construct.
The introduction provides a good background for investigation of the construct of “social intolerance” in the context of common mental health issues during the pandemic. However, the literature on the impact of additional factors, which are set to be the study variables, such as social support and specific socio-demographic data (race, house-sharing, etc.) is missing. Please support with the previous research variables that were selected for the investigation.
Authors' response: Thank you for this comment, we have added paragraphs to the introduction to support the investigated associations.
Results
-Please provide a table with descriptive statistics on demographic data and the study measurements. For example. It is not clear from your report who “youngest individuals” are or what “lower schooling” means.
Authors' response: Thank you for this comment, we have added table 1 with descriptive statistics of sociodemographic characteristics and levels of stress, intolerance to uncertainty, receiving and giving social support. We insert information regarding these estimates in the methods section.
-Furthermore, the descriptive presentation of the data should be supported with statistical analysis. For example, χ2 may provide a good estimation whether the differences in the CMD prevail between the young and older participants are not occasional.
Authors' response: We are grateful for this comment, we have added to table 1 the p-value obtained by Pearson's chi-square test and information regarding this estimate in the methods section.
Discussion
-Line 355: Please provide a reference for the statement. What new information presents the following sentence starting with “high prevalence values of sadness/depression…” (Line 358)?
Authors' response: Thank you for this comment, the references have been added, and we have added a sentence at the end of the paragraph about the relationship between the feelings experienced by men and the high prevalence of CMD among them.
-Lines 361-366: Please move this paragraph to the Introduction. Be oriented toward explanation of your findings.
Authors' response: Thanks for this comment, the above paragraph has been moved to the introduction.
-In general, the discussion is too long with multiple repetitions on the main topics, such as vulnerability due to intolerance to uncertainty (lines 453-461) or cultural issues (Lines 472-477/ 373-379).
Authors' response: Thank you for this comment, we have chosen to delete these paragraphs.
-In addition, some pieces of information are not relevant for the study, even though they are generally interesting. For example, the statements on non-biological nature of mental health (“These results reinforce that mental health goes beyond the biological issues, …”) of the not fit the study purposes and design since no biological measurements were done.
Authors' response: Thank you for this comment, we chose to delete the paragraph where this sentence was located.
-I recommend to revise and reorganize the discussion to bring the main information without duplications and with a focus on the research main concepts.
Authors' response: We appreciate this comment, we believe that with the changes suggested by the reviewer and made by us, the text became more concise and highlighted the main concepts of the research.
-In general, the authors bring a lot of references, while part of information still remains ungrounded. Please review the references and use the stronger ones.
Authors' response: We are grateful for this comment, we believe that by excluding content that was not directly related to the study's findings, we also reduced the excess of references, leaving only those considered essential by us in the text.

Round 2
Reviewer 1 Report
First of all, thanks for the possibility to review the manuscript,
It’s a good work, the authors have listened my suggestions
With that, the article has improved its writing and understanding
In summary, the manuscript can be published in its current form,
I encourage the authors to continue working in this line.
My congratulations